

# Screening and identification of endometrial proteins as novel potential biomarkers for repeated implantation failure

Chong Wang[1,2], Ying Feng[1], Wen-Jing Zhou[1], Zhao-Jun Cheng[1], Mei-Yan Jiang[1], Yan Zhou[1] and Xiao-Yang Fei[1]

[1] Hangzhou Women's Hospital (Hangzhou Maternity and Child Health Care Hospital), Hangzhou, China
[2] Affiliated Hangzhou First People's Hospital, Zhejiang University School of Medicine, Hangzhou, China

## ABSTRACT

Inadequate endometrial receptivity may be responsible for the low implantation rate of transferred embryos in in vitro fertilization (IVF) treatments. Patients with repeated implantation failure (RIF) impact the clinical pregnancy rate for IVF. We collected endometrial tissue during the implantation window of hysteroscopy biopsies from September 2016 to December 2019 and clinical data were collected simultaneously. Patients were divided into RIF and pregnant controls group according to pregnancy outcomes. A total of 82 differentially expressed endometrial proteins were identified, including 55 up-regulated proteins ($>1.50$-fold, $P < 0.05$) and 27 down-regulated proteins ($<0.67$-fold, $P < 0.05$) by iTRAQ labeling coupled with the 2D LC MS/MS technique in the RIF group. String analysis found interactions between these proteins which assembled in two bunches: ribosomal proteins and blood homeostasis proteins. The most significant enriched Gene Ontology terms were negative regulation of hydrolase activity, blood microparticle, and enzyme inhibitor activity. Our results emphasized the corticosteroid-binding globulin and fetuin-A as the specific proteins of endometrial receptivity by Western-blot. Our study provided experimental data to establish the objective indicator of endometrial receptivity, and also provided new insight into the pathogenesis of RIF.

## INTRODUCTION

According to the World Health Organization, infertility is most prevalent in South Asia, Sub-Saharan Africa, North Africa/Middle East, Central/Eastern Europe, and Central Asia (*Mascarenhas et al., 2012*). There is a 25% infertility rate among couples of childbearing age in China (*Zhou et al., 2018*). Assisted reproductive technology (ART) is currently the most effective method to assist infertile patients (*Kushnir et al., 2017*).

In vitro fertilization and embryo transfer (IVF-ET) are the most common methods of ART. Despite the success of these methods, some patients still have difficulty becoming pregnant even after multiple transplantations (2–6 times) of high-quality embryos. These patients are classified as repeated implantation failure (RIF) patients (*Margalioth et al.,*

Corresponding author
Xiao-Yang Fei, feixy1962@163.com

*2006*) and impact the clinical pregnancy rate of ART. Embryo quality, which can be evaluated using several criteria, and endometrial receptivity are the two key factors for successful implantation. Endometrial receptivity is defined as the period during which the endometrial epithelium acquires a functional, but transient, ovarian steroid-dependent status that supports blastocyst acceptance and implantation. This period is called the window of implantation (WOI). Approximately 25.9% of IVF-ET cases have a displaced WOI (*Ruiz-Alonso et al., 2013*), and the lack of synchronization between the embryo and endometrial receptivity may be one of the causes of RIF. The standard for evaluating endometrial receptivity is the pinopodes in endometrial histology but these can change depending on the sample used and the period in the menstrual cycle (*Acosta et al., 2000*). Ultrasound examination is a widely used, noninvasive and inexpensive test, and includes endometrial thickness, endometrial type, endometrial volume, and uterine artery and sub-endometrial blood flow. However, ultrasound examination has little ability to predict the pregnancy rate with strong subjectivity. A number of studies have applied omics techniques to analyze the human endometrium along different menstrual cycles, using biopsy or curettage (*Bissonnette et al., 2016*; *Chen et al., 2009*; *DeSouza et al., 2005*; *Parmar et al., 2009*; *Rai et al., 2010*; *Ruiz-Alonso, Blesa & Simón, 2012*; *Yap et al., 2011*). Previous research (*Bissonnette et al., 2016*; *Chen et al., 2009*; *DeSouza et al., 2005*; *Parmar et al., 2009*; *Rai et al., 2010*) used proteomic techniques as 2D differentials in-gel electrophoresis (DIGE), Nanobore LC-MS/MS, and MALDI-TOF-TOF to study the endometrium protein changes between the proliferative and secretory phase. *Yap et al. (2011)* identified IL-11 regulated plasma membrane proteins ANXA2, and the lipid-raft protein FLOT1 in human endometrial epithelial cells in vitro in the receptive phase.

We screened differential WOI endometrial proteins using iTRAQ labeling coupled with a 2D LC-MS/MS technique to find potential biomarkers for RIF patients. Our study provided experimental data to establish an objective indicator of endometrial receptivity and a new way to reveal the pathogenesis of RIF.

## MATERIALS & METHODS

### Sample collection

This study was approved by the Ethics Committee of the Faculty of Medicine (Hangzhou Women's Hospital, China) (Approval Number: 2016001-10). Written informed consent was obtained from all subjects before endometrial collection.

The workflow of our study is shown in Fig. 1. Failure of three or more cycles in which reasonably high-quality embryos were transferred was defined as RIF (*Margalioth et al., 2006*). Grade I and grade II embryos were determined to be high-quality embryos (*Gardner et al., 1998*). The blastocyst quality was determined according to the definitions by *Gardner et al. (1998)*. Blastocysts were considered high-quality if they had a grade 3 or 4 blastocoel, a grade A or B inner cell mass, and a grade A or B trophectoderm on days 5 or 6. We collected data from fifty-two RIF cases and 135 pregnancy cases undergoing IVF-ET treatment at our hospital between September 2016 to December 2019. Data including age, follow-up outcomes, and clinical examination findings were collected. The endometrium

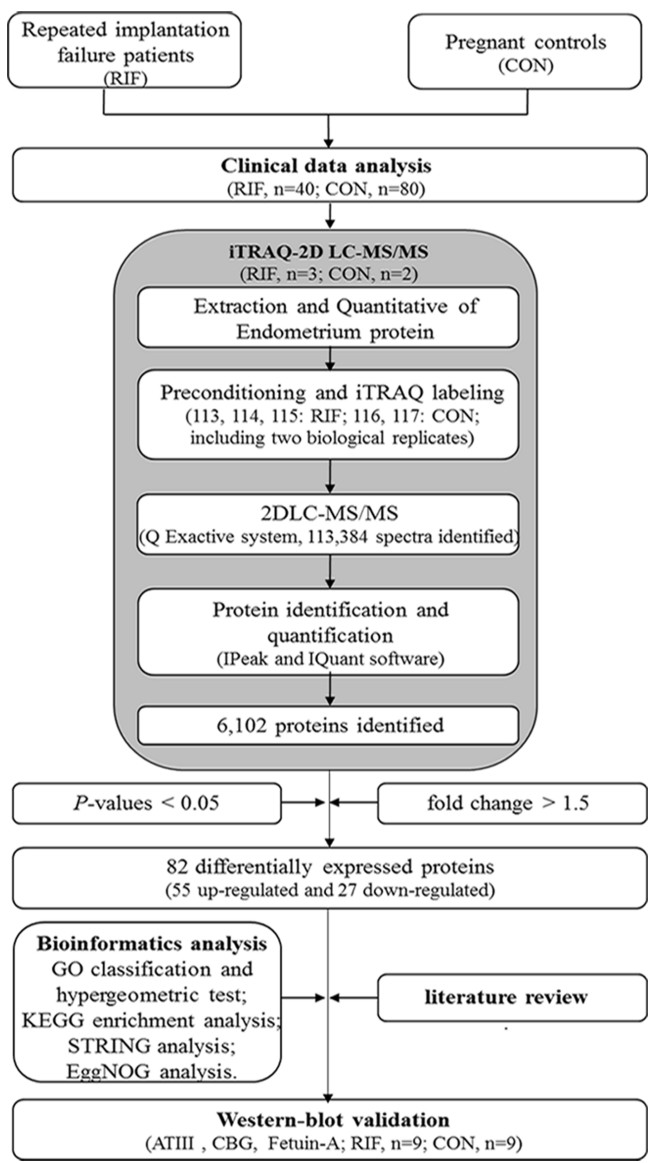

**Figure 1** **The workflow for endometrial biomarkers of repeated implantation failure (RIF) and pregnant controls (CON).**

was collected at WOI by hysteroscopy biopsy at LH+7. The endometrium was then washed with saline immediately and frozen in liquid nitrogen and stored until protein extraction.

## Endometrium protein extraction

We randomly selected 2 pregnancy cases and 3 RIF cases for endometrial protein extraction. Protein extraction was performed using a lysis buffer after grinding the sample (100 mg) to powder in liquid nitrogen. Phenylmethanesulfonyl fluoride (PMSF) was added to a final concentration of 1mM and ethylene diamine tetraacetic acid (EDTA) was added to a final concentration of 2mM, held for 5 min, and then dithiothreitol (DTT) was added to a final

concentration of 10mM. The sample was placed in an ice bath ultrasound for 5 min and the lysate was centrifuged at 15,000 g for 20 min. The supernatant was precipitated 5 times with 1 mL acetone and incubated at 20 °C for 2 h and then centrifuged at 15,000 g at 4 °C for 20 min. The precipitate was washed in chilled acetone, incubated at 20 °C for 30 min, and centrifuged again at 15,000 g and 4 °C for 20 min. The washing was repeated twice, then air dried and the precipitate was re-dissolved with the lysis buffer. The precipitate was centrifuged again at 15,000 g and 4 °C for 20 min after an ice bath ultrasound for 5 min. The supernatant was determined using the Bradford Protein Assay Kit to detect the protein concentration.

## Protein digestion and iTRAQ-2D LC-MS/MS

A solution of 1:50 trypsin (Promega, USA)-to-protein mass was prepared. Then, a total of 100 μg of protein from each group was digested with prepared solution at 37 °C for 16 h. The peptides were reconstituted in 0.2M TEAB and processed according to the manufacturer's protocol for 8-plex iTRAQ reagent (AB SCIEX, Framingham, MA, USA). Three biological replicates of the RIF group were labeled with 113, 114, and 115 isobaric tags, respectively. The peptides with two biological replicates from the pregnant group were labeled with 116 and 117 isobaric tags, respectively.

A high pH reversed-phase chromatography column (Phenomenex, Gemini-NX 3u C18110A, 150*2.00 mm) was used for the first-dimensional fractionation procedure. We collected 16 fractions in all, and then dried them for next LC-MS analysis. The fractions were re-suspended in 2% acetonitrile containing 0.1% formic acid, and then loaded into a C18 trap column (Acclaim PepMap 75 μm ×150 mm, C18, 3 μm, 100A). Then, online chromatography separation was performed on the nanoLC system (Dionex Ultimate 3000 RSLCnano) (*Fan, Wang & Wang, 2019*). The trapping and desalting procedures were carried out at a flow rate of 3 μL/min for 5 min with 100% solvent A (0.1% formic acid, 2% acet- onitrile and 98% water). The peptides were eluted using a 65 min gradient of buffer A (0.1% formic acid) to buffer B (80% ACN containing 0.1% formic acid) at 300 nL/min on an analytical column (Acclaim PepMap 75 μm ×15 cm C18-CL, 3 μm 100 Å, Thermo160321). Q Exactive system (Thermo Scientific) fitted with a Nanospray ion source was used to acquire tandem MS data. Specific steps are as follows: data were acquired using an ion spray voltage of 2.2 kV. MS spectra across the scan range of 350–1,800 m/z with a 70,000 resolution using maximum injection time (60 ms) per spectrum. Twenty of the most intense precursors per MS cycle were selected for fragmentation and were detected with 100 ms maximum injection time. Tandem mass spectra were recorded at a 17,500 resolution with the rolling collision energy turned on and iTRAQ reagent collision energy adjustment turned on. The lock mass option was enabled for more accurate measurements. Dynamic exclusion was set for 10 s.

Finally, the acquired MS/MS data were analyzed using IPeak and IQuant software as former researchers (*Wen et al., 2015*; *Wen et al., 2014*; *Fan, Wang & Wang, 2019*). Only proteins identified at global FDR $\leq 1\%$ with $\geq 1$ peptide were considered for further downstream analysis. A differentially expressed protein was determine only if it is identified and quantified with at least one significant peptide with the $P < 0.05$ and fold change $> 1.5$.

## Bioinformatics analysis

Principal components analysis (PCA) was performed to confirm the sample repeatability. Functional annotation was performed using the Gene Ontology (GO) database (http://www.geneontology.org) and included the cellular component, molecular function, and biological process. The differentially expressed protein–protein network was analyzed by STRING software (http://www.string-db.org/). The Kyoto Encyclopedia of Genes and Genomes (KEGG) database (http://www.genome.jp/kegg/ or http://www.kegg.jp/) was used to predict the main metabolic pathways (*Kanehisa et al., 2007*). We obtained the significantly enriched GO/pathway items by hypergeometric test. The EggNOG database (http://eggnogdb.embl.de) was used for pairwise orthology predictions, functional annotation, and classification (*Huerta-Cepas et al., 2015*).

## Western-blot analysis

We lysed endometrial tissues with 200 μ L of RIPA lysate (P0013B, Beyotime, Shanghai, China) plus 1 mM PMSF at 4 °C for 30 min, and then harvested the supernatant with centrifugation at 11,000 g for 10 min. The harvested protein concentrations were measured with a BCA quantitative kit (P0009, Beyotime). Samples were subjected to polyacrylamide gel electrophoresis, and were transferred onto a PVDF membrane (IPVH00010, Millipore, Massachusetts, USA). The membrane was blocked by 5% skimmed milk powder solution at room temperature for 2 h and was incubated with primary antibodies, including antithrombin III (rabbit monoclonal, ab126598, abcam, Cambridge, United Kingdom), cortisol binding globulin (rabbit monoclonal, ab110648, abcam), fetuin-A (alpha-2-HS-glycoprotein, rabbit monoclonal, ab137125, abcam), GAPDH (mouse monoclonal, 60004-1-Ig, proteintech, Beijing, China), and alpha tubulin (rabbit polyclonal, 11224-1-AP, proteintech) at 4 °C overnight. Secondary antibodies as goat anti-mouse IgG-HRP (BK0023, BEST, Xian, China) and goat anti-rabbit IgG-HRP (BK0027, BEST) were then incubated with membrane at room temperature for 1.5 h. The blots were visualized using the ECL Plus Luminous Kit (S17851, Yeasen, Shanghai, China). At last, the results were measured with Image J software.

## Statistical analysis

Parametric data were tested using the chi-square test for the composition ratios and t-tests for means of two groups. Nonparametric analysis was carried out using the Mann–Whitney U-test. Parametric data were presented as mean $\pm$ SD while nonparametric data were presented as median $\pm$ IQR, and $P < 0.05$ was considered to be statistically significant by the SPSS software, version 16.0 (SPSS, Chicago, IL). Our clinical data was able to identify significant differences in 81.45% of RIF cases and controls at a statistical support level of $\alpha = 0.05$ with a $d = 0.5$ applying a one tail model calculated by Gpower 3.0.5.

## RESULTS

### Clinical data analysis

We recruited 52 RIF patients and 135 pregnant patients undergoing IVF-ET treatment between September 2016 to December 2019. Data, including age, follow-up outcomes, and

**Table 1 General and clinical data of repeated implantation failure patients (RIF) and pregnant controls.**

| | RIF group ($n = 40$) | Control group ($n = 80$) |
|---|---|---|
| Age (year)[a] | 32.60 ± 3.90 | 32.60 ± 3.84 |
| Infertility years (year)[a] | 4.32 ± 2.49 | 3.88 ± 2.84 |
| Infertility type (n)[b] | | |
| Oviduct factors | 20 | 33 |
| Ovulation disorders | 7 | 17 |
| Oarium factors | 3 | 9 |
| Pelvic cavity factors | 3 | 6 |
| Male factors | 3 | 7 |
| Unknown causes | 4 | 8 |
| Endometrium thickness of transplant day (mm)[a] | 9.71 ± 1.77 | 10.52 ± 2.27 |
| High-quality rate of transplant embryo (%)[b] | 87.75 (222/253) | 90.91 (140/154) |
| BMI (kg/m2)[a] | 21.07 ± 2.85 | 21.26 ± 2.55 |
| AMH (ng/mL)[a] | 3.59 ± 2.55 | 3.24 ± 2.13 |
| D-dimer (mg/L)[c] | 220.00 ± 202.50 | 210.00 ± 230.00 |
| Fibrinogen (G/L)[a] | 2.49 ± 0.65 | 2.54 ± 0.72 |
| Basal hormone level | | |
| Follicle-stimulating hormone (IU/L)[a] | 5.30 ± 2.37 | 5.26 ± 2.13 |
| Estradiol (pg/mL)[c] | 27.00 ± 17.30 | 25.00 ± 22.50 |
| Progesterone (ng/mL)[a] | 0.62 ± 0.29 | 0.65 ± 0.52 |
| Prolactin (ng/mL)[a] | 14.02 ± 6.18 | 13.87 ± 6.46 |
| Luteinizing hormone (IU/L)[a] | 3.14 ± 1.83 | 2.93 ± 2.15 |
| Testosterone (ng/mL)[a] | 0.53 ± 0.43 | 0.46 ± 0.26 |
| Transformation day hormone level | | |
| Estradiol (pg/mL)[a] | 554.33 ± 268.31 | 585.78 ± 398.97 |
| Progesterone (ng/mL)[a] | 0.48 ± 0.27 | 0.52 ± 0.34 |

**Notes.**
All data are presented as the mean ± SD.
BMI, body mass index; AMH, anti-Mullerian hormone.
[a] $P$-value between two groups using the $t$-test.
[b] $P$-value between two groups using the chi-square test.
[c] $P$-value between two groups using the Mann–Whitney U-test.

clinical examination findings were collated into databases. After correcting for age, since age may affect the pregnancy rate, a total of 40 subjects with RIF under 40 years old were paired with 80 pregnant subjects by age, body mass index, and treatment time and then analyzed. There were no significant differences between RIF patients and pregnant controls in general and clinical data ($P > 0.05$, Table 1).

## Endometrial proteomics results

We identified a total of 6,102 proteins through iTRAQ-2D LC-MS/MS from 113,384 spectra and 31,024 peptides, respectively (Table S1). Among the 6,102 identified proteins, 5,840 had GO annotations (95.71% of all proteins); 5,504 had KEGG annotations (90.20% of all proteins); and 6,097 had EggNOG annotations (99.92% of all proteins).

We performed quality control on the quantitative results with volcano maps and the distributions of coefficient of variation. We selected the fold change for 6,102 proteins and found 285 proteins with fold change in RIF cases/pregnant controls >1.5 or <0.67. And we only used proteins with $P$ <0.05. Further screening revealed 82 differentially expressed proteins in RIF patients compared with the pregnant controls, including 55 up-regulated proteins (>1.50-fold, $P < 0.05$) and 27 down-regulated proteins (<0.67-fold, $P < 0.05$) (Table 2). The hierarchical clustering provided a visualized mode to display the clustering patterns of the differentially expressed proteins between the groups (Fig. 2).

## Bioinformatics analysis results

Gene Ontology analysis of differentially expressed proteins revealed that most of the proteins were involved in the response to stimulus (42 proteins), extracellular region (35 proteins), and structural molecule activity (10 proteins) (Fig. 3A). The most significantly enriched GOs were negative regulation of hydrolase activity, blood microparticle, and enzyme inhibitor activity through hypergeometric testing (Fig. 3B). Seven proteins (SPB6, APOA1, GMIP, THBG, CBG, ANT3, and FETUA) were identified in the hydrolase activity term, seven proteins (VTDB, IGHG4, APOA1, A1AG2, FETUA, ANT3, and A1AG1) were identified in the blood microparticle term, and another seven proteins (ANT3, FETUA, CBG, ASPN, SPB6, THBG, and APOA1) were identified in the enzyme inhibitor activity term. Among these proteins, ANT3 and FETUA were identified in prior studies (*Hannan et al., 2010*; *DeSouza et al., 2005*) which played an important molecular function in endopeptidase inhibitor activity by GO analysis (Fig. 3C). String analysis found interactions between these proteins (Fig. 3D) which assembled in two bunches: ribosomal proteins and blood homeostasis proteins.

In addition, the KEGG pathway mapping revealed the immune system (seven proteins), transport and catabolism (five proteins), and translation (five proteins) pathways (Fig. 4A). Enriched KEGG pathway analysis showed the ribosome and primary immunodeficiency pathways as significant with $P$ <0.05 (Fig. 4B). Finally, we used the EggNOG database to determine that the differential proteins associated with RIF are mostly clustered in classifications including posttranslational modification, protein turnover, chaperones (22 proteins), translation, ribosomal structure and biogenesis (eight proteins), and carbohydrate transport and metabolism (six proteins) (Fig. 4C).

## Western-blot results

We verified endometrial antihrombin-III (ANT3, P01008), corticosteroid-binding globulin (CBG, P08185), and fetuin-A (FETUA, P02765) levels using the Western-blot. We found significantly higher levels of CBG and fetuin-A in RIF patients (Fig. 5). A significant difference in CBG and fetuin-A was found in RIF patients using grayscale detection by Image J (1.39 fold, $P = 0.003$; 1.47 fold, $P = 0.002$; respectively).

## DISCUSSION

Assisted reproductive technology has made rapid progress over the last 40 years. However, the clinical pregnancy rate still only ranges between 33.8 and 42.7% (*ESHRE et al., 2016*;

**Table 2  Differentially expressed proteins and their expression levels quantified by iTRAQ-2DLC-MS/MS.**

| Protein ID | Alternative name | Protein name | iTRAQ ratio |
|---|---|---|---|
| **Increased in RIF/Controls** | | | |
| Q9NYZ3 | GTSE1 | G2 and S phase-expressed protein 1 | 3.42 |
| Q9UN19 | DAPP1 | Dual adapter for phosphotyrosine and 3-phosphotyrosine and 3-phosphoinositide | 3.38 |
| Q5W111 | SPRY7 | SPRY domain-containing protein 7 | 3.11 |
| P50225 | ST1A1 | Sulfotransferase 1A1 | 3.06 |
| Q86UB9 | TM135 | Transmembrane protein 135 | 2.59 |
| P05230 | FGF1 | Fibroblast growth factor 1 | 2.56 |
| Q8NFU3 | TSTD1 | Thiosulfate:glutathionesulfurtransferase | 2.49 |
| Q8NDA2 | HMCN2 | Hemicentin-2 | 2.49 |
| Q9H477 | RBSK | Ribokinase | 2.40 |
| P15169 | CBPN | Carboxypeptidase N catalytic chain | 2.37 |
| P02763 | A1AG1 | Alpha-1-acid glycoprotein 1 | 2.15 |
| Q86VY4 | TSYL5 | Testis-specific Y-encoded-like protein 5 | 2.03 |
| Q9UKJ8 | ADA21 | Disintegrin and metalloproteinase domain-containing protein 21 | 1.98 |
| P02794 | FRIH | Ferritin heavy chain | 1.91 |
| Q96RG2 | PASK | PAS domain-containing serine/threonine-protein kinase | 1.89 |
| P08294 | SODE | Extracellular superoxide dismutase [Cu-Zn] | 1.88 |
| P08582 | TRFM | Melanotransferrin | 1.87 |
| Q9P2H3 | IFT80 | Intraflagellar transport protein 80 homolog | 1.82 |
| A0A0B4J1U7 | 1U7|HV601 | Immunoglobulin heavy variable 6-1 | 1.81 |
| P02765 | FETUA | Alpha-2-HS-glycoprotein | 1.78 |
| Q9HCJ0 | TNR6C | Trinucleotide repeat-containing gene 6C protein | 1.78 |
| P01008 | ANT3 | Antithrombin-III | 1.77 |
| Q14353 | GAMT | Guanidinoacetate N-methyltransferase | 1.77 |
| P19652 | A1AG2 | Alpha-1-acid glycoprotein 2 | 1.77 |
| Q86X19 | TMM17 | Transmembrane protein 17 | 1.75 |
| O76041 | NEBL | Nebulette | 1.74 |
| Q03167 | TGBR3 | Transforming growth factor beta receptor type 3 | 1.70 |
| Q96EX3 | WDR34 | WD repeat-containing protein 34 | 1.67 |
| P02792 | FRIL | Ferritin light chain | 1.66 |
| Q9NWK9 | BCD1 | Box C/D snoRNA protein 1 | 1.64 |
| P15559 | NQO1 | NAD(P)H dehydrogenase [quinone] 1 | 1.64 |
| Q96C11 | FGGY | FGGY carbohydrate kinase domain-containing protein | 1.64 |
| Q99598 | TSNAX | Translin-associated protein X | 1.63 |
| Q9NZM6 | PK2L2 | Polycystic kidney disease 2-like 2 protein | 1.62 |
| P08185 | CBG | Corticosteroid-binding globulin | 1.62 |
| Q9BXN1 | ASPN | Asporin | 1.61 |
| Q15063 | POSTN | Periostin | 1.61 |

**Table 2** (*continued*)

| Protein ID | Alternative name | Protein name | iTRAQ ratio |
|---|---|---|---|
| Q9NVP4 | DZAN1 | Double zinc ribbon and ankyrin repeat-containing protein 1 | 1.60 |
| Q14651 | PLSI | Plastin-1 | 1.57 |
| P02774 | VTDB | Vitamin D-binding protein | 1.57 |
| P05543 | THBG | Thyroxine-binding globulin | 1.57 |
| P25311 | ZA2G | Zinc-alpha-2-glycoprotein | 1.56 |
| Q9NZJ9 | NUDT4 | Diphosphoinositol polyphosphate phosphohydrolase 2 | 1.55 |
| Q9H649 | NSUN3 | tRNA (cytosine(34)-C(5))-methyltransferase, mitochondrial | 1.55 |
| P80108 | PHLD | Phosphatidylinositol-glycan-specific phospholipase D | 1.55 |
| Q96AB6 | NTAN1 | Protein N-terminal asparagine amidohydrolase | 1.54 |
| Q9H9L4 | KANL2 | KAT8 regulatory NSL complex subunit 2 | 1.54 |
| P02647 | APOA1 | Apolipoprotein A-I | 1.53 |
| P48509 | CD151 | CD151 antigen | 1.53 |
| Q9UBW7 | ZMYM2 | Zinc finger MYM-type protein 2 | 1.52 |
| Q96LD8 | SENP8 | Sentrin-specific protease 8 | 1.52 |
| P35237 | SPB6 | Serpin B6 | 1.52 |
| Q5HYK9 | ZN667 | Zinc finger protein 667 | 1.51 |
| Q99735 | MGST2 | Microsomal glutathione S-transferase 2 | 1.51 |
| Q9NQG6 | MID51 | Mitochondrial dynamics protein MID51 | 1.50 |
| **Decreased in RIF/Controls** | | | |
| Q9H9C1 | SPE39 | Spermatogenesis-defective protein 39 homolog | 0.27 |
| O43314 | VIP2 | Inositol hexakisphosphate and diphosphoinositol-pentakisphosphate kinase 2 | 0.36 |
| P28908 | TNR8 | Tumor necrosis factor receptor superfamily member 8 | 0.40 |
| Q96RD9 | FCRL5 | Fc receptor-like protein 5 | 0.44 |
| P62805 | H4 | Histone H4 | 0.47 |
| Q9P107 | GMIP | GEM-interacting protein | 0.48 |
| Q96S82 | UBL7 | Ubiquitin-like protein 7 | 0.53 |
| P18124 | RL7 | 60S ribosomal protein L7 | 0.53 |
| P56202 | CATW | Cathepsin W | 0.53 |
| Q6P179 | ERAP2 | Endoplasmic reticulum aminopeptidase 2 | 0.57 |
| Q05086 | UBE3A | Ubiquitin-protein ligase E3A | 0.58 |
| P43403 | ZAP70 | Tyrosine-protein kinase ZAP-70 | 0.59 |
| P46781 | RS9 | 40S ribosomal protein S9 | 0.59 |
| Q8TAF3 | WDR48 | WD repeat-containing protein 48 | 0.60 |
| Q8N4H5 | TOM5 | Mitochondrial import receptor subunit TOM5 homolog | 0.60 |
| P08729 | K2C7 | Keratin, type II cytoskeletal 7 | 0.61 |
| Q07020 | RL18 | 60S ribosomal protein L18 | 0.62 |
| Q9BRX8 | F213A | Redox-regulatory protein FAM213A | 0.62 |
| P07197 | NFM | Neurofilament medium polypeptide | 0.62 |
| P15954 | COX7C | Cytochrome c oxidase subunit 7C, mitochondrial | 0.62 |

**Table 2** (*continued*)

| Protein ID | Alternative name | Protein name | iTRAQ ratio |
|---|---|---|---|
| P01861 | IGHG4 | Immunoglobulin heavy constant gamma 4 | 0.64 |
| P61313 | RL15 | 60S ribosomal protein L15 | 0.64 |
| Q96SI1 | KCD15 | BTB/POZ domain-containing protein KCTD15 | 0.65 |
| P07196 | NFL | Neurofilament light polypeptide | 0.65 |
| Q3SX64 | OD3L2 | Outer dense fiber protein 3-like protein 2 | 0.65 |
| Q02543 | RL18A | 60S ribosomal protein L18a | 0.65 |
| Q8NGY6 | OR6N2 | Olfactory receptor 6N2 | 0.66 |

*Sunderam et al., 2019*). After the implementation of blastocyst transplantation, the clinical pregnancy rate increased to 60.4% (*Ozgur et al., 2018*). RIF is the key factor affecting the pregnancy rate in IVF. Embryo quality, uterine factors (uterine cavity lesions, adenomyosis, endometrial receptivity, etc.), immune factors (embryo immunity, maternal immunity), and a multifactor effect could all lead to RIF but inadequate endometrial receptivity is the major cause of decreased pregnancy success in RIF patients. We collected the endometrium of IVF-ET patients during WOI and divided the samples into the RIF group and pregnant control group according to pregnancy outcomes. Specific proteins related to endometrial receptivity were screened using iTRAQ-2D LC-MS/MS.

Through iTRAQ-2D LC-MS/MS and bioinformatics analysis, 82 differential proteins were obtained in the endometrium of RIF patients during the WOI, of which 55 were higher ($> 1.50$ times, $P < 0.05$) in RIF patients and 27 were lower ($< 0.67$ times, $P < 0.05$) in RIF patients. The differential proteins obtained in this study have also been identified in the early proteomic studies. Hannan et al. obtained seven differential proteins of the uterine lavage fluid of pregnant/non-pregnant patients in the WOI by 2D-DiGE. The antithrombin III (ANT3, P01008) was significantly increased in non-pregnant patients by immunohistochemistry (*Hannan et al., 2010*). We found that the expression of ANT3 was 1.77 times as much as that in the pregnant group, which was similar to previous results, indicating the reliability of iTRAQ-2D LC-MS/MS. The results for ANT3, alpha-1-acid glycoprotein 1 (P02763), vitamin D-binding protein (P02774), and FETUA (P02765) were also consistent with previous studies on the menstrual phase (*DeSouza et al., 2005*). Ribosomal proteins and apolipoproteins have also been identified in ours and other previous studies (*DeSouza et al., 2005*; *Domínguez et al., 2009*; *Pérez-Debén et al., 2019*).

The differentially expressed proteins were related to the immune system and primary immunodeficiency revealing by KEGG analysis and KEGG enrichment analysis (Figs. 3A, 3B). Therefore, the change of immune response in RIF patients is self-evident. It has been shown that the abnormal and functional defects of immune cells and molecules in endometrium during implantation can lead to pregnancy failure (*Liu et al., 2016*). Therefore, we suspect that the change of the endometrial immune microenvironment may lead to RIF, and may lead to a better clinical treatment for RIF patients.

Our study revealed that most of the differentially expressed proteins were annotated with "posttranslational modification, protein turnover, chaperones" and "translation, ribosomal structure and biogenesis function" based on the EggNOG database, except for

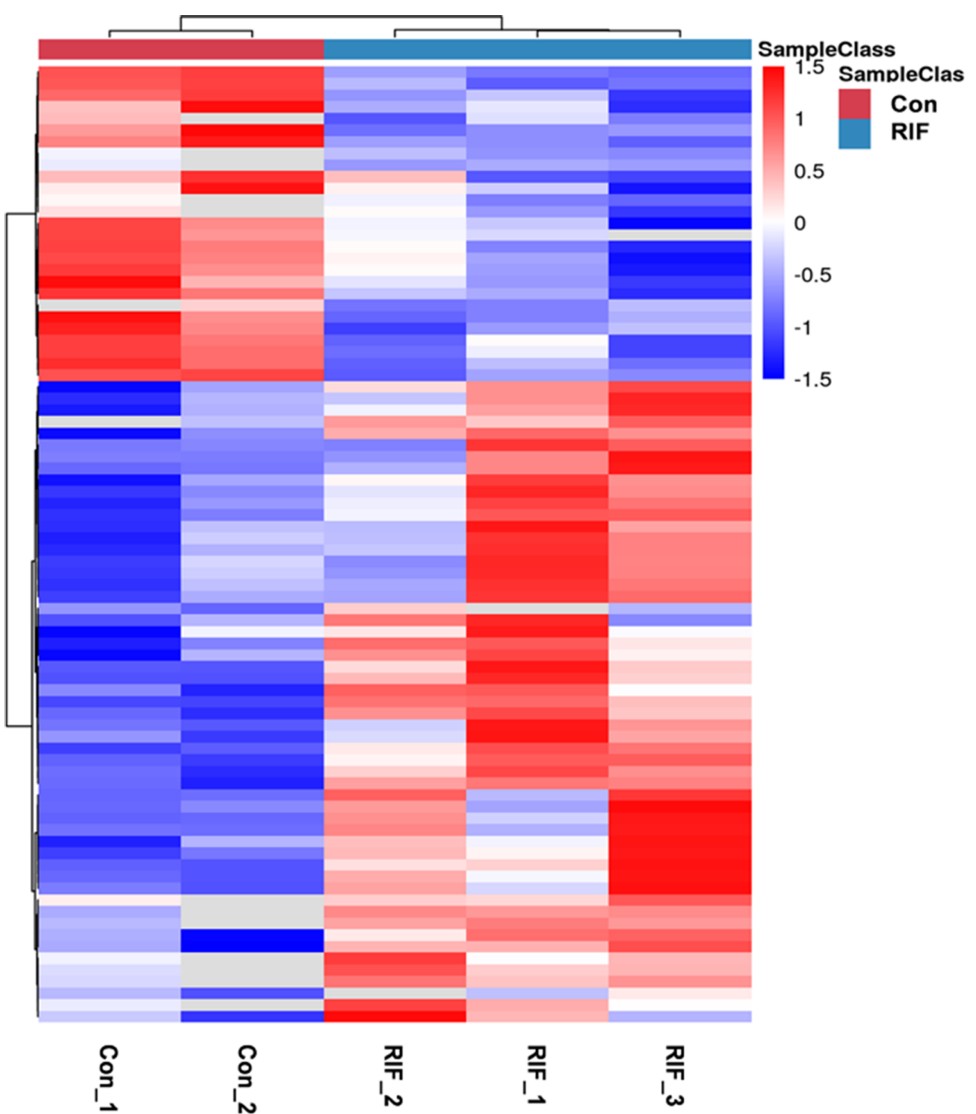

**Figure 2** **The hierarchical clustering for endometrium proteins between repeated implantation failure (RIF) and pregnant controls (Con) groups.** The red color showed the up-regulated expression, and the blue color represented the down-regulated expression. The color from red/blue to white represented the ratio from large to small.

those with ''function unknown''. So, we have reason to believe that modification after translation, synthesis and degradation, folding, maintenance, intracellular transport, and mRNA translation might be the key functions changed in embryo implantation. The results of enriched KEGG and String analysis in our study also confirmed that translation was impacted.

The proteins in the most significantly enriched GOs contained ANT3 and FETUA, which was consistent with results from previous studies (*DeSouza et al., 2005*; *Hannan et al., 2010*). ANT3 was also related to the immune system as highlighted in the KEGG analysis. These proteins were selected as candidates for validation. Protein was also selected

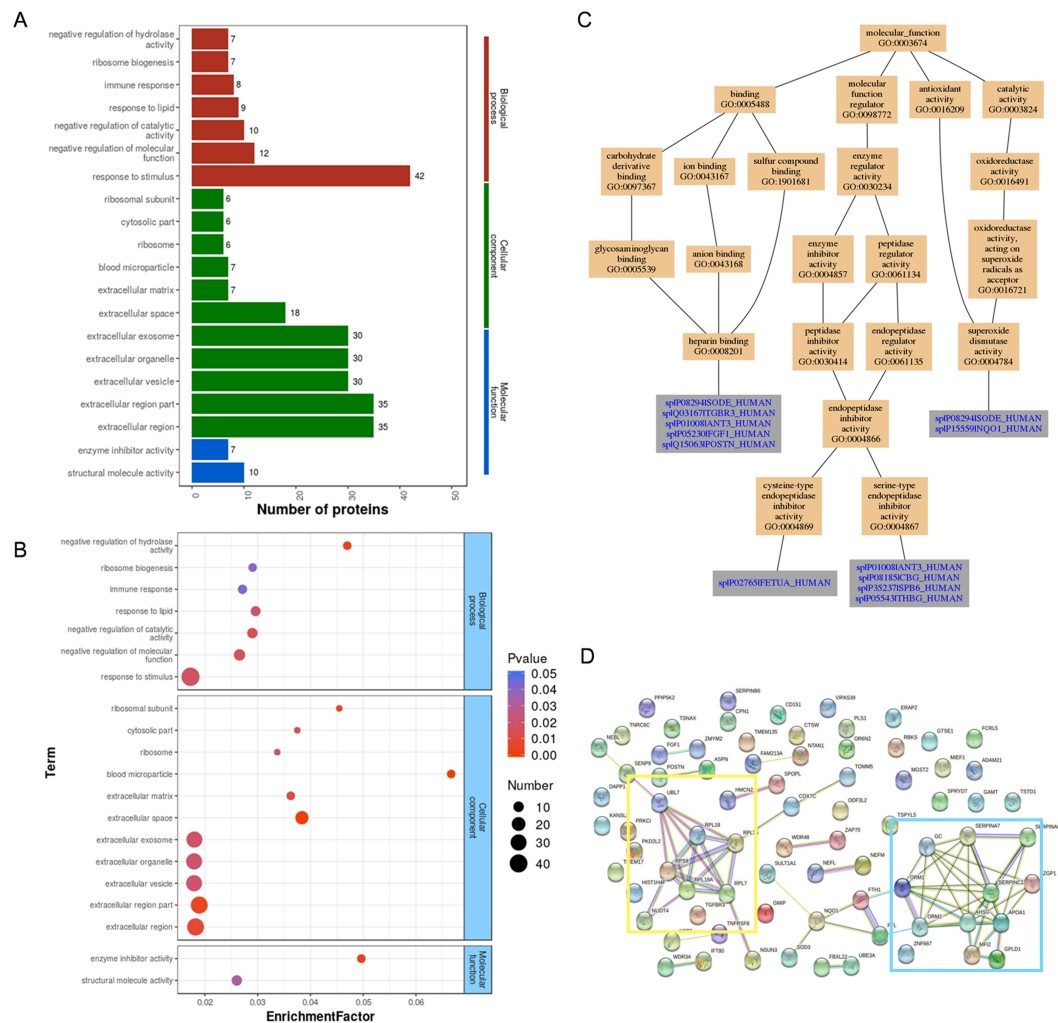

**Figure 3 GO analysis and String analysis of the set of endometrium proteins biomarker candidates for repeated implantation failure.** (A) GO analysis of 82 differentially expressed proteins revealed seven significant GO terms in biological process, 11 significant GO terms in cellular component, and two significant GO terms in molecular function ($P < 0.05$). (B) The enrichment analysis revealed 20 significant GO terms by hypergeometric test ($P < 0.05$). GO terms with bigger enrichment factor indicate the greater degree of enrichment. (C) Molecular function of GO terms for proteins identified both in our and prior studies. (D) Network nodes represent proteins while edges represent protein-protein associations which were already known (light blue and purple) or predicted (other colors) by String analysis. Proteins enclosed in color-coded outlines are mainly involved in ribosomal proteins (yellow) and blood homeostasis proteins (blue).

based on a review of the literature as CBG. *Misao et al. (1995)* suggested that the decrease of progesterone level in the blood can lead to an increased CBG expression level in the endometrium. Low progesterone levels may lead to higher miscarriage rates and lower live birth rates in frozen embryo transfer patients (*Gaggiotti-Marre et al., 2019*).

Antithrombin-III is a representative protein of the prethrombotic state, which is encoded by ANT3. The prethrombotic state is thought to be a major cause of RIF (*Qublan*

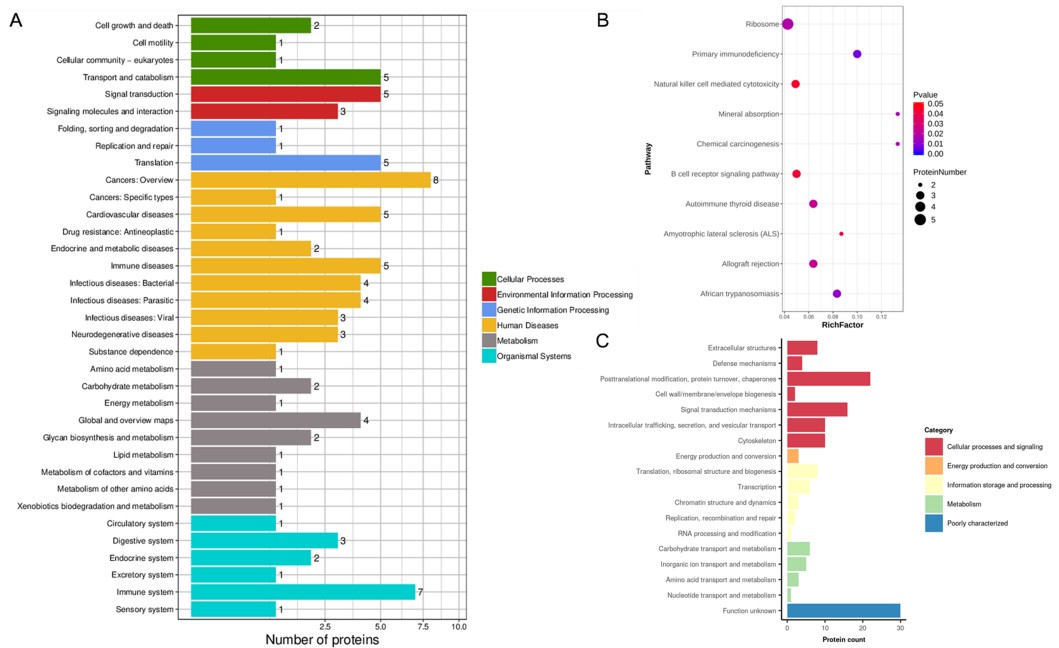

**Figure 4 KEGG analysis and EggNOG analysis of the set of endometrium proteins biomarker candidates for repeated implantation failure.** (A) KEGG pathway analysis of 82 differentially expressed proteins using KEGG database revealed 35 pathways. (B) The enrichment analysis of KEGG pathways revealed 10 significant pathways by hypergeometric test ($P < 0.05$). Pathways with bigger rich factor indicate the greater degree of enrichment. (C) EggNOG analysis predicted pairwise orthology and functional classification for 82 differentially expressed proteins.

*et al., 2006*). Our proteomics results corroborated those of *Hannan et al. (2010)*, which revealed up-regulated levels of antithrombin III in non-pregnant patients (*Hannan et al., 2010*). However, antithrombin III showed no significant difference in RIF patients and pregnant controls. These differences may be due to the differences in the validation tests used. We used the quantitative Western-blot for validation while Hannan et al. used immunohistochemical localization. Therefore, antithrombin III may not change in RIF patients.

Corticosteroid-binding globulin was a multifaceted component in cortisol delivery, also in acute and chronic inflammation, and metabolism and neurocognitive function (*Meyer et al., 2016*). The increased CBG level during pregnancy was important at the materno-fetal interface (*Lei et al., 2015*). We found significantly higher levels of CBG in RIF patients (Fig. 4). *Misao et al. (1995)* suggested that the decrease of progesterone levels in the blood may increase CBG expression in the endometrium. However, we found a lower level of serum progesterone in RIF patients, but the difference was not significant (Table 1) and a larger sample of research may be needed. The endometrial CBG content is thought to originate in the plasma (*Kreitmann, Derache & Bayard, 1978*), thus, RIF patients may have a positive outcome with supplemental progesterone administration.

Fetuin-A is defined as the inhibitor of ectopic calcification in circulation, which also takes part in multiple metabolic pathways such as insulin resistance, vascular calcification,

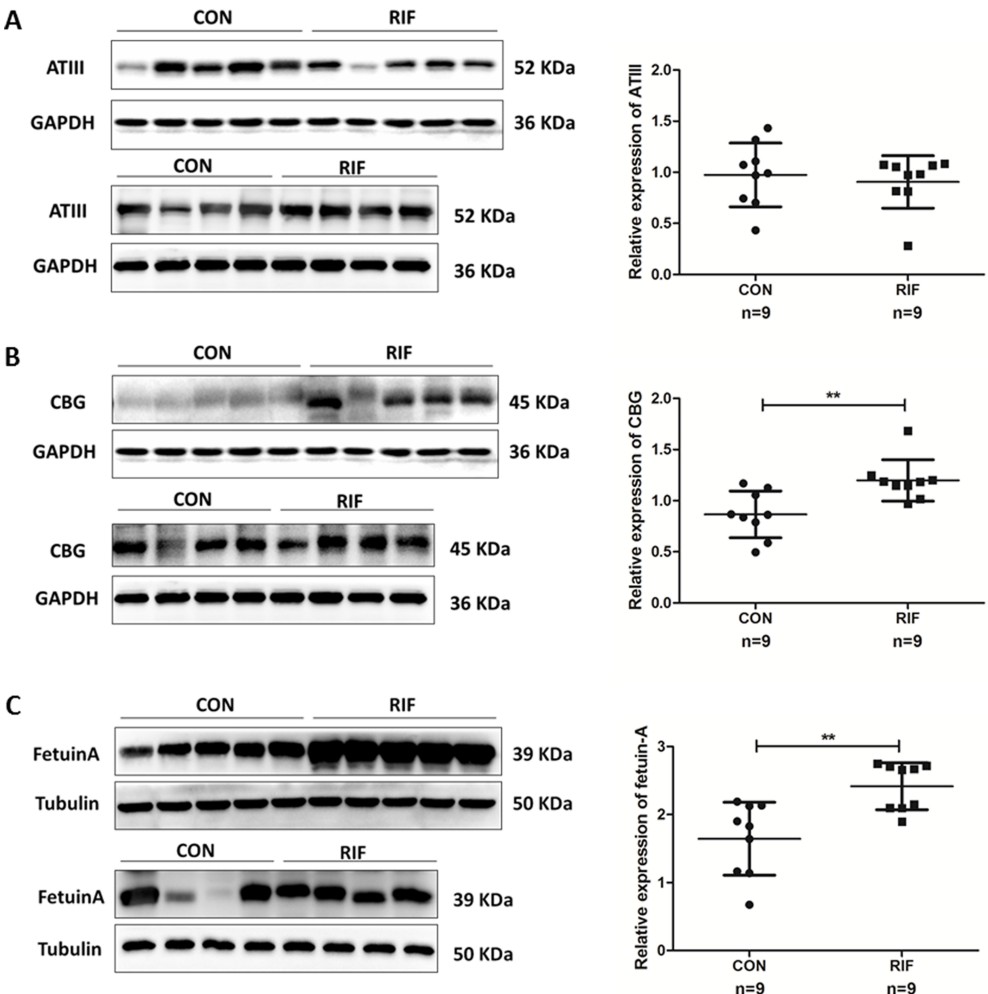

**Figure 5** **Proteins expression levels change between repeated implantation failure (RIF) and pregnant controls (CON).** (A) antithrombin III (ATIII) (52 KDa); (B) corticosteroid-binding globulin (CBG) (45 KDa); and (C) fetuin-A (39 KDa) were analyzed by western-blot in nine cases of pregnant controls (CON) and repeated implantation failure (RIF) patients. Beside each western-blot picture, grayscale analysis is represented where the intensity of each protein band is compared to a GAPDH/tubulin band. Grayscale analysis showed a similar tendency to the iTRAQ analysis, with a higher protein abundance of CBG in RIF cases (1.39 fold, $P = 0.003$), also higher fetuin-A (1.47 fold, $P = 0.002$).

and inflammation (*Bilgir et al., 2010*; *Ishibashi et al., 2010*; *Wang & Sama, 2012*). Fetuin-A is also popular in studies on adverse pregnancy outcomes like pre-eclampsia (*Sanhal et al., 2016*) and gestational diabetes mellitus (*Kansu-Celik et al., 2019*). A previous study on pre-eclampsia concluded that fetuin-A may decrease trophoblast viability and invasion caused by the inhibition of receptor tyrosine kinase activity (*Gomez et al., 2012*). Our study supported the results that showed that elevated fetuin-A may lead to failed implantation and cause an adverse pregnancy outcome. Meanwhile, *Ozgu-Erdinc et al. (2020)* demonstrated that serum fetuin-A level were also increased in implantation failure patients in IVF cycles.

Additional studies on the regulation of the level of fetuin-A as a treatment strategy may improve implantation success.

## CONCLUSIONS

We screened the endometrial proteomics of RIF patients using iTRAQ-2D LC-MS/MS at WOI, revealing that the endometrial immune microenvironment may lead to RIF. Our validated results confirmed CBG and fetuin-A as the specific protein for RIF patients. Our results provide experimental data to establish the objective indicator of endometrial receptivity and give a new insight into the pathogenesis of RIF.

## ACKNOWLEDGEMENTS

We wish to thank the patients who participated in our study.

### Funding

This work was supported by the Medicine and Health Science Technology Project of Zhejiang province, China (Grant number 2017KY550) and the Zhejiang Provincial Natural Science Foundation of China (grant number LQ18H040009). The funders had no role in study design, data collection and analysis, decision to publish, or preparation of the manuscript.

### Grant Disclosures

The following grant information was disclosed by the authors:
The Medicine and Health Science Technology Project of Zhejiang Province, China: 2017KY550.
The Zhejiang Provincial Natural Science Foundation of China: LQ18H040009.

### Competing Interests

The authors declare there are no competing interests.

### Author Contributions

- Chong Wang conceived and designed the experiments, analyzed the data, authored or reviewed drafts of the paper, and approved the final draft.
- Ying Feng and Wen-Jing Zhou analyzed the data, prepared figures and/or tables, and approved the final draft.
- Zhao-Jun Cheng performed the experiments, analyzed the data, prepared figures and/or tables, and approved the final draft.
- Mei-Yan Jiang and Yan Zhou performed the experiments, prepared figures and/or tables, and approved the final draft.
- Xiao-Yang Fei conceived and designed the experiments, authored or reviewed drafts of the paper, and approved the final draft.

## Human Ethics

The following information was supplied relating to ethical approvals (i.e., approving body and any reference numbers):

This study was approved by the Ethics Committee of the Faculty of Medicine (Hangzhou Women's Hospital, China) (Ethical Application Ref: 2016001-10).

## Data Availability

Data is available at ProteomeXchange Consortium (http://proteomecentral. proteomexchange.org) via the iProX partner repository, accession numbers: PXD021068 .

Clinical data are available in the Supplemental File.

## Supplemental Information

Supplemental information for this article can be found online at http://dx.doi.org/10.7717/ peerj.11009#supplemental-information.

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
