# Peer review of "Screening and identification of endometrial proteins as novel potential biomarkers for repeated implantation failure"

_PeerJ, doi:10.7717/peerj.11009_

## Round 0.1 · original submission · Major Revisions

All three expert reviewers find this manuscript interesting and important for understanding repeated implantation failure. However, language needs major refinement, and lots of details need to be added to clarify the experimental design and data analysis/conclusion. Authors may keep the current data and sample size without adding more samples, but have to clearly explain or add relevant discussion if a certain sample size seems not abundant. See reviewers’ comments for more details.

·

Basic reporting

A very worthy topic which merits publication as papers in this field are generally lacking.

I find the syntax and general English usage to be a bit incoherent and feel the document, while obviously translated from the native language, could benefit from further work by a native English speaker.
Line 57 onward. The proteomic references are incomplete and should include the work of Alonso and J Horchadas on the European continent.

The selection of the candidate proteins out of 6000 plus proteins should be fully elucidated. Should more WB's of other candidates been performed to make the argument more convincing?
The reviewer is not clear why CBG selected above other candidates present in a cluster, is there an introduction of Bias in this selection?

Experimental design

I'm a little bit disappointed with the number of patients selected for this pilot study, I feel the numbers could be higher to allow more robust conclusions. I also note some Male factor patients in the RIF group and wonder why these patients did not make inclusion into the "control" group.
Are controls used by merit of the fact that they happened to get pregnant while part of the IVF program. They may have had conditions not representative of true population controls which may thus influence the results.
Is it possible to aquire endometrial samples for this protocol which are not related to IVF in any way?

Use of LC MS and WB is a nice way to definitively assess the proteins involved rather than the heretofore accepted methods of mRNA evaluations, ultrasound, immunohistochemistry and flow cytometry. In this aspect the paper is unique and merits publication.

Ln78. The WOI should be defined specifically and how it was evaluated. e.g. LH+7. Clarity should be given that WOI may be variable in approx 30% of cases and this may be the origin of the RIF.
Ln82. Do I take this as patients were batch processed rather than processed individually?
Fig 1. it is unclear what Control 1 and 2 and RIF 1, 2 and 3 are..are these individuals selected as examples?
All figure legends need to fully fleshed out with explanatory annotations.

Validity of the findings

Repeat implantation failure is by no means limited only to endometrial receptivity and other factors, other than the rather obvious embryo quality, which lead to this negative outcome should be explored fully to fully enlighten the reader. Examples would be a "hostile" immune profile.

Could normalization of the CBG expression profile be demonstrated with supplemental progesterone administration?

I think the conclusions could be more fully fleshed out and the clinical implications discussed

Additional comments

Meritorious work and well conducted using suitable techniques. I'd like to see better use of English if possible and more coherent arguments used for selection of the protein candidates used for WB and ultimately decided as being directly involved or otherwise in the process of RIF

Reviewer 2 ·

Basic reporting

no comment

Experimental design

Specific comments:
In the abstract, we have no idea of the compared groups with the proteomic approach
Line 20: what do you mean by 'poor endometrial receptivity ?
Lines 56-57: insufficient literature (PMID:15602768, PMID:19714818, PMID:18793766, PMID:21137015, PMID: 26760977...)
Lines 78: What is the exact timing for hysteroscopybiopsy. How was determine the WOI ? How much time after the LH surge or ovulation was performed the hysteroscobiopsy?
In the general and clinical data of RIF patients (n=40) and pregnant groups (n=80), some data presented as the mean ± SD are issued from too small clinical data. For example, the mean for endometrial thickness the day of embryo transfer included only 3 patients in each group (RIF and pregnant group); therefore, these data are not representative of the cohort.
In the same way, there are only 28/40 and 55/80 patients for the AMH concentration in the RIF and pregnant group respectively, etc. Moreover, I do not find certain values of means ± SD using the clinical database provided in the excel file (eg. [P] = 0.61 ± 0.88 ng/ml in the RIF group , and not 0.47 ± 0.27 ng/ml).
Do you perform a normality test before choosing a parametric test ?
The number of samples used for protein extraction is very low, only 2 for the pregnant group and 3 for the RIF group. We recommend increasing the number of samples per group.
The total number of peak detected must be supplied.
How much peaks as significantly differentially expressed were identified between the 2 groups ?
The differences in peaks intensity volume must be supplied
how much peaks were selected for protein identification ? all peaks between 350 to 1800 m/z?
How much samples used for the first-dimensional fractionation procedure ? Please, clarify
How was selected candidates for validation by western blot ?
How to explain that very few protein candidates were consistent with prior studies ?
A workflow for the protein detection and identification will be appreciate
Lines 161-162: as some clinical data are not representative of the entire cohort, it cannot be said that there were no significant different in clinical data between groups
In the western blot results, bands must be quantified and fold changes provided.
Fig. 2D, unreadable
In the fig. 3, I'm not sure that KEGG pathways comprising only 1 protein were really significant pathways. It is not relevant. Same remark for the EggNOG analysis.
Lines 216-218: Not totally agree with this comment as Domínguez et al identified the 40S ribosomal protein SA and the L-Plastin proteins, while in the present study the 60S ribosomal protein L18 and Plastin-1 were identified.

Validity of the findings

no comment

Additional comments

The manuscript of Wang et al. analyses the difference in endometrial protein expression profile during the implantation window between pregnant vs. RIF patients using the iTRAQ labeling coupled with 2D LC MS/MS technique. The idea is not novel and several articles have already been published in this field using omic technologies. However, this is an interesting article that may have potential implications in the understanding of molecular mechanisms associated with implantation failure. However, experiments and experimental design were not presented in a satisfactory manner to judge the present study. In addition, the literature referred is incomplete and the authors should conduct a more careful review of the data. Therefore, there are a number of items that should be addressed to improve the quality of the manuscript (see specific comments).

Reviewer 3 ·

Basic reporting

The manuscript needs to be thoroughly revised for the appropriate use of the English language. Word choice and unclear sentences make the text difficult to read at times. Words, expressions, and sentences that should be revised have been highlighted on the PDF version of the manuscript, which is attached to this review. Two supplementary files were provided in a language other than English; one of them seems to have an English version. Figure 4 legend should be improved to provide more information. There is no indication of sample size. Otherwise, the manuscript meets the expected journal's criteria.

Experimental design

The authors have clearly stated a relevant and meaningful research question, as the lack of clinically relevant markers of improved endometrial receptivity is a major limitation in the field. Methods are well-described and adequate to answer the proposed question.

Validity of the findings

The question addressed by the authors is not novel, but it still remains relevant. The data are robust and statistically sound; however, the conclusion, although liked to the main question, is not supported by the results. The authors report the CBG protein as a novel and objective indicator of endometrial receptivity, which has not been shown by the data. Substantial experimentation should be performed to validate these results. Given the relatively large sample size, authors could try to predict RIF according to CBG protein expression, for example, which would provide further support to the observed relationship.

Additional comments

The manuscript entitled "Screening and identification of endometrium proteins as novel potential biomarkers for repeated implantation failure" used endometrial biopsy samples of RIF and pregnant patients and a proteomic approach to try to identify endometrial markers of receptivity, and validated two identified candidates by western blotting. The authors propose that Cortisol Binding Globin is a marker of endometrial receptivity. Overall, the manuscript has a relevant question and an appropriate experimental approach; however, it needs a thorough language review and needs to revise its conclusion, which is not fully supported by the data. In its current format, the manuscript is not adequate for publication.

Annotated reviews are not available for download in order to protect the identity of reviewers who chose to remain anonymous.

---

## Round 0.2 · Minor Revisions

Both expert reviewers have confirmed the significant improvement of the manuscript, however, some questions still need to be addressed or explained, although extra experiments seem not to be needed at this step.

·

Basic reporting

N/A

Experimental design

N/A

Validity of the findings

N/A

Additional comments

Thanks you for the revisions, I think they have clarified the content and improved the paper.

Reviewer 3 ·

Basic reporting

The use of professional English was much improved. Few observations made in that regard.

Please revise:

Throughout the text, authors use abbreviations to begin sentences; Is that adequate?
L149 - skimmed milk powder
L232-233 - specify "higher" and "lower" than what?

Experimental design

L146 - use g force rather than rpm;
L160-161 - how can any of the variables from the dataset be analyzed by a paired t-test? As far as I can tell those are not paired samples;
L170-176 - What is the purpose of comparing the clinical data of the 120 patients between RIF and pregnant individuals? Considering that molecular data are compared within a subset of samples, it seems more useful to know how groups compare within each subset of samples;
L212-216 - It would significantly increase the power and impact of the data if Western blot was performed on all available samples;

Validity of the findings

The study's conclusions are still not adequate. There is not a functional experiment and the large dataset was not explored to validate the differentially expressed markers. There is excessive speculation being used to make a conclusion.

Additional comments

The manuscript was significantly improved; however, major issues still remain to be addressed before the study can be considered for publication.

---

## Round 0.3 · accepted · Accept

All minor questions have been well addressed.